# An acidic microenvironment in Tuberculosis increases extracellular matrix degradation by regulating macrophage inflammatory responses

**Ashley M. Whittington** [1]*, **Frances S. Turner**[2], **Friedrich Baark**[1], **Sam Templeman**[1], **Daniela E. Kirwan**[3], **Candice Roufosse**[1], **Nitya Krishnan**[1], **Brian D. Robertson**[1], **Deborah L. W. Chong**[3], **Joanna C. Porter**[4], **Robert H. Gilman**[5], **Jon S. Friedland**[3]

1 Department of Infectious Disease, Imperial College London, London, United Kingdom, 2 Edinburgh Genomics, University of Edinburgh, Edinburgh, United Kingdom, 3 Institute of Infection and Immunity, St. George's, University of London, London, United Kingdom, 4 Centre for Inflammation & Tissue Repair, Respiratory Medicine, University College London, London, United Kingdom, 5 Department of International Health, Johns Hopkins University, Baltimore, Maryland, United States of America

* a.whittington@nhs.net

**Data Availability Statement:** The raw sequencing data has been deposited in the European

## Abstract

Mycobacterium tuberculosis (M.tb) infection causes marked tissue inflammation leading to lung destruction and morbidity. The inflammatory extracellular microenvironment is acidic, however the effect of this acidosis on the immune response to M.tb is unknown. Using RNA-seq we show that acidosis produces system level transcriptional change in M.tb infected human macrophages regulating almost 4000 genes. Acidosis specifically upregulated extracellular matrix (ECM) degradation pathways with increased expression of Matrix metalloproteinases (MMPs) which mediate lung destruction in Tuberculosis. Macrophage MMP-1 and -3 secretion was increased by acidosis in a cellular model. Acidosis markedly suppresses several cytokines central to control of M.tb infection including TNF-α and IFN-γ. Murine studies demonstrated expression of known acidosis signaling G-protein coupled receptors OGR-1 and TDAG-8 in Tuberculosis which are shown to mediate the immune effects of decreased pH. Receptors were then demonstrated to be expressed in patients with TB lymphadenitis. Collectively, our findings show that an acidic microenvironment modulates immune function to reduce protective inflammatory responses and increase extracellular matrix degradation in Tuberculosis. Acidosis receptors are therefore potential targets for host directed therapy in patients.

## Author summary

Tuberculosis (TB) is one of the leading causes of death from a single infectious disease. Infection with *Mycobacterium tuberculosis* generates an immune response which causes inflammation. In some patients this inflammation controls infection but in those that develop TB this inflammation causes extensive lung tissue destruction resulting in

Nucleotide Archive (ENA) under accession number PRJEB20229 and is publicly available. Other data tables are included as supplementary information.

**Funding:** This work was funded by awards to AMW and JSF from the Medical Research Council (MR/L001683/1), National Institute of Health and Care Research (ACF-2010-21-032) and Mason Medical Research Trust. BDR and NK were supported by the Imperial College London Trust. The funders had no role in study design, data collection and analysis, decision to publish, or preparation of the manuscript.

**Competing interests:** The authors have declared that no competing interests exist.

mortality and morbidity. Lung destruction arises due to excessive production of a group of enzymes, the matrix metalloproteinases (MMPs), which degrades lung tissue. Understanding which factors promote protective, rather than tissue damaging, inflammation will identify new potential treatments. Inflammation causes tissue affected by TB to become more acidic. In this study we found that this acidity profoundly alters the immune response of infected cells to promote production of MMPs and suppresses production of key proteins required to control infection. This acidity is detected by receptors on the surface of TB infected cells (TDAG-8 and OGR-1). Drugs which target these receptors may therefore be of therapeutic benefit to help control disease and reduce tissue damage in TB.

## Introduction

*Mycobacterium tuberculosis* (M.tb) is an intracellular pathogen of macrophages. It infects a third of the world's population resulting in 1.6 million deaths annually [1]. Tuberculosis deaths are rising due to the COVID pandemic [2]. In the majority of infected individuals who remain well, disease is prevented by activation of macrophage intracellular defense mechanisms including phagolysosome maturation, generation of reactive oxygen species, apoptosis and autophagy which contains M.tb [3, 4]. Failure to contain infection results in clinical Tuberculosis (TB), a disease characterized by high levels of proinflammatory cytokines and extensive tissue inflammation. Inflammation leads to tissue destruction causing morbidity and facilitates mycobacterial growth and transmission [5]. Lung destruction in TB is immune-mediated, resulting from the activity of enzymes particularly Matrix Metalloproteinases (MMPs). MMPs degrade pulmonary extracellular matrix (ECM) and are derived from both M.tb-infected macrophages, and lung parenchymal cells stimulated by proinflammatory cytokines. MMPs-1,-3,-8, -9 and -10 are all increased in TB patients [6–8] with collagenase MMP-1 the key driver of collagen breakdown and tissue necrosis [9, 10]. The balance of MMP and cytokine activity is therefore critical to disease outcomes in TB.

Inflammation disrupts tissue homeostasis; generating a distinct inflammatory microenvironment characterized by hypoxia [11], tissue acidosis [12] with elevated extracellular lactate concentrations [13] and fever range temperature [14]. An acidic microenvironment arises during the host response to infection due to lactic acid secretion by inflammatory cells, and inflamed tissues typically have an extracellular pH around 7.0 compared to pH 7.4 in healthy tissue. In TB patients, CSF [15], pleural [16], peritoneal [17] and lung lining fluid [18] are more acidic than healthy tissue. Extracellular acidosis is signaled by the Ovarian Cancer G-Protein Coupled Receptor (OGR) family of cell surface receptors, including OGR-1, T-cell death associated gene-8 (TDAG-8) and G-protein coupled receptor-4 (GPR-4) [19]. These receptors utilize conserved histidine residues to detect extracellular protons and are silent at pH 7.8 and maximally active at pH 6.8 [20]. TDAG-8 is predominantly expressed in lymphoid tissue [21] and TDAG-8 knock out mice have enhanced delayed type hypersensitivity responses [22]. OGR-1 expression is widespread and OGR1 deficient mice have attenuated pulmonary [23] and intestinal [24] inflammatory responses indicating that acidosis receptors may have immune function.

Extracellular acidosis has diverse effects on LPS-stimulated non-human primary macrophages and macrophage cell lines including reduced TNF-α secretion, which requires TDAG-8 signalling ([25]), and increased inflamasome activation [26], chemotaxis [27], phagocytosis [28] and NFκB signalling [29]. In contrast, TNF-α secretion by primary human monocyte derived macrophages (MDMs), a major cell type infected by M.tb, is unchanged by acidosis

[30]. Unstimulated human macrophage cell lines respond differently to primary cells with extracellular acidosis suppressing COX-2 and IL-6 levels in immortalised cells but increasing levels in primary cells.

We therefore investigated for the first time how extracellular acidosis and lactate in the inflammatory microenvironment regulates primary human macrophage responses to M.tb. RNA-seq of uninfected macrophages demonstrated limited transcriptional changes with acidosis with activation of cellular stress response gene sets. M.tb-infected MDMs show marked transcriptional change in acidosis which enhances expression of gene sets regulating ECM degradation and glucose metabolism. Further investigations in a cellular model confirmed acidosis increases M.tb driven MMP-1 secretion and suppresses TNF-α. TDAG-8 is highly expressed in granulomas and on multinucleate giant cells in a murine model of TB infection and regulates MMP-1 activity in a human cellular model.We further demonstrate that TDAG-8 is highly expressed in patients with active TB. Extracellular lactate enhances MMP-1 secretion independently of acidosis. The acidic inflammatory microenvironment in TB is therefore immunomodulatory driving matrix degradation and inhibiting activity of key protective cytokines.

## Results

### Extracellular acidosis drives transcriptional change in human macrophages which is synergistically increased by M.tb stimulation

Firstly, uninfected human MDMs cultured at either neutral pH 7.4 or in acidosis at pH 7.0 were RNA-sequenced to investigate the effects of acidosis alone. RNA-seq was then performed on MDMs infected with live virulent M.tb at pH 7.4 to determine the effects of infection, and on MDMs infected and cultured at pH 7.0 to assess the combined effects of infection and acidosis. Human MDMs were generated by monocyte culture in M-CSF alone so MDMs were not polarised prior to stimulation [31]. General trends in the resulting data were investigated with Principal Component Analysis (PCA). Plotting of the first and second principal components shows samples cluster by both M.tb infection state and pH suggesting both stimuli trigger independent transcriptional responses (Fig 1A). Three biological replicates cluster closely on analysis of the third principal component with one (Donor 2) having a somewhat different pattern of gene expression (S1A Fig). As experimental conditions, RNA integrity and sequencing quality control were similar across all samples, this donor was included in further analysis as this level of diversity likely reflects the heterogeneity of human responses to M.tb. Hierarchical clustering of the 50 most differentially expressed genes shows grouping of samples by M.tb infection and clustering of infected samples by pH, further demonstrating distinct transcription signatures for M.tb infection and acidosis (Fig 1B).

Differential gene expression analysis was performed between conditions as shown in Fig 1C Full gene lists in S1 Dataset. For the first time, the effect of extracellular acidosis alone on human macrophages was dissected by comparing gene expression of unstimulated MDMs at pH 7.0 with unstimulated MDMs at pH 7.4 (analysis labelled Effect of Acidosis (uninfected)). Acidosis alone significantly upregulated expression of 178 genes and downregulated 37 genes. Comparison of Mtb infected with uninfected MDMs cultured at pH 7.4 (Analysis labelled Effect of Infection (pH7.4)) demonstrated infection drives greater transcriptional change than acidosis alone with upregulation of 1556 and downregulation of 1441 genes. Comparing M.tb-infected MDMs with control cells, both cultured at pH 7.0, shows the effect of M.tb infection at pH 7.0. Infection at pH 7.0 greatly increased transcriptional change with 2919 up- and 2616 down-regulated genes. Thus, 3102 genes (1611 upregulated, 1491 downregulated) are differentially expressed by Mtb infection at pH 7.0 but not by infection at pH 7.4 (Fig 1D). Finally, the

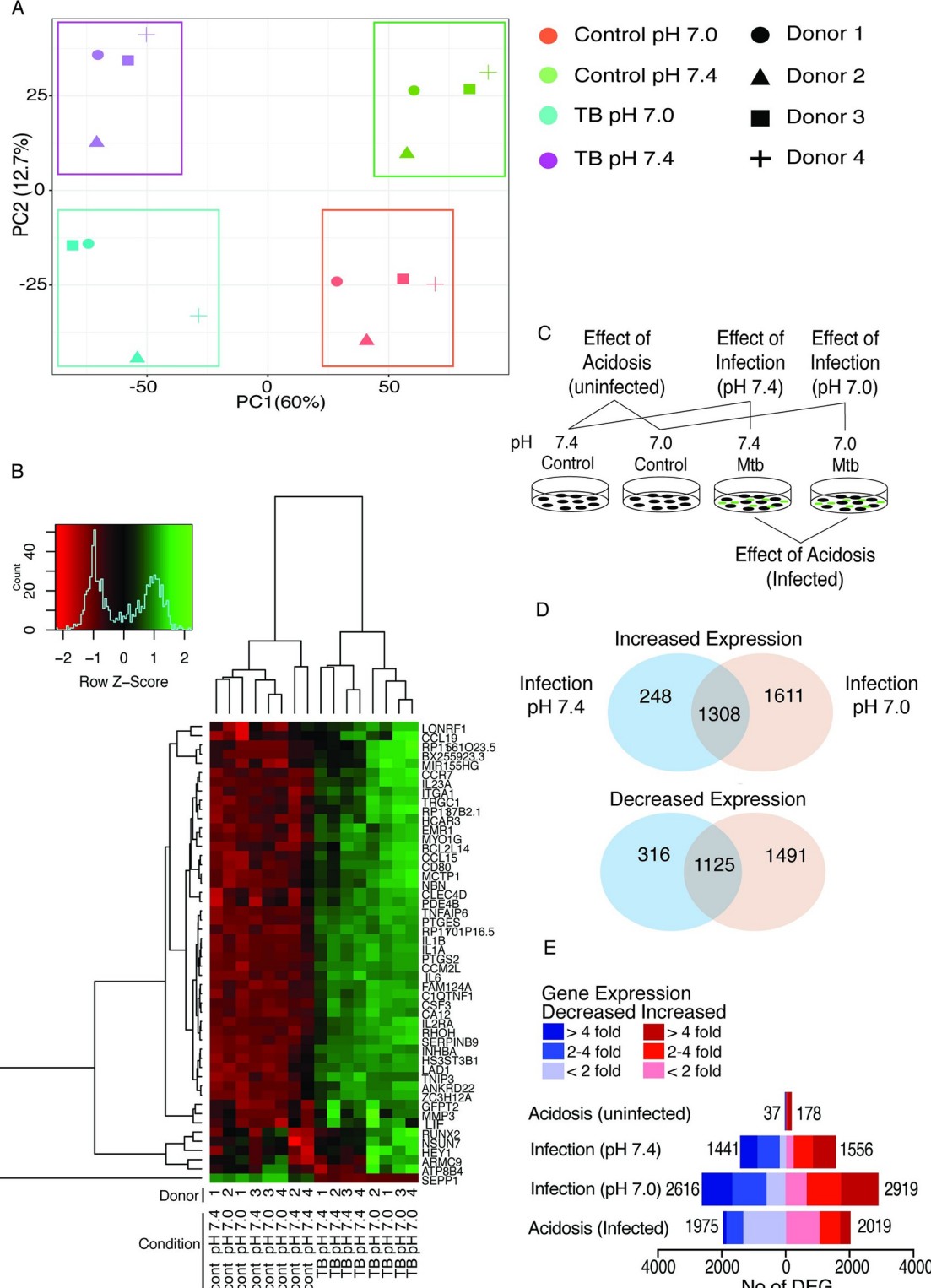

**Fig 1. Transcriptional change induced by acidosis and M.tb infection in macrophages.** RNA-seq was performed on control and M.tb infected MDMs incubated at pH 7.4 or acidic pH 7.0. (A) Principal Component Analysis (PCA) showing the first and second principal component on the x and y axis respectively. Total variance attributable to that PC indicated. (B) Clustered expression intensities standardised to z-score are shown for the 50 most differentially expressed genes (by *p* value) across all samples. Each sample is labelled with condition, pH and replicate number (C) Schematic of the conditions compared in differential expression

analysis. (D) Venn diagram showing overlap of differentially expressed genes between the effect of infection at pH 7.4 and infection at pH 7.0 (E) Magnitude and number of differential gene expression in each analysis. Light bars <2 fold, medium shading 2–4 fold, dark bars >4 fold.

overall effect of acidosis on infected MDMs was determined by directly comparing gene expression of infected MDMs at pH 7.0 with M.tb infected MDMs at pH 7.4 (Effect of Acidosis (Infected)). In this analysis 2019 and 1975 genes were up and downregulated respectively, representing DE of 30.7% of the genome expressed by control macrophages. The magnitude of fold change across these DE genes is significant with 437 genes differently expressed by >4 fold (Fig 1E). Smear plots showing the magnitude of differential expression of individual genes for each analysis are in S1C Fig. Therefore, although extracellular acidosis has some effect on MDM gene expression alone, acidosis has a marked synergistic interaction with M.tb to induce significant transcriptional change in macrophages.

## Extracellular Acidosis downregulated inflammatory pathways in unstimulated macrophages

Acidosis alone in uninfected MDMs causes differential expression of 210 genes (178 up, 37 down) representing only 1.6% of total expressed genes. Genes for chemokines (CXCR4, CXCR7, CXCL5 AND CXCL15), signal transducer STAT4 and IL-1 receptor (IL1R1) were the only genes upregulated by acidosis with direct immune function. The biological effects of acidosis alone were determined by performing functional analysis using Gene Set Enrichment Analysis (GSEA) on the results of the Acidosis (uninfected) analysis. GSEA revealed no Reactome gene sets upregulated by acidosis alone. A total of 219 gene sets were downregulated, many highly significantly with the top 133 of 219 having FDR-q values <0.001. GSEA outputs using the Reactome Knowledgebase gene sets [32] were similar to Gene Ontology (GO) database results and complete lists of enriched gene sets in each analysis are shown in S2 Dataset. Classification by uppermost category of the Reactome event hierarchy revealed downregulated gene sets are involved in the cell cycle, gene expression, DNA repair and replication, specific disease states, signal transduction and metabolism (Fig 2A) reminiscent of the transcriptional change seen with other forms of cellular stress [33]. Twelve immune system gene sets were downregulated including NF-κB activation pathways, antigen processing and cross presentation, Endoplasmic reticulum-phagosome pathways and C-type lectin signalling pathways (Fig 2B) suggesting acidosis is anti-inflammatory in uninfected macrophages. Regulators of MMP secretion in TB including MAP Kinase signalling [34], Cell-ECM [35] and Basigin [36] interactions were downregulated as were metabolic pathways including the Tricarboxylic Acid Cycle (TCA cycle) and the electron transport chain (Fig 2C).

## M.tb infection upregulates multiple inflammatory pathways and MMP genes

We next assessed the effect of M.tb infection at pH 7.4. In the Infection (pH 7.4) analysis numerous cytokine and chemokine genes of known relevance in TB immunity were highly upregulated. The top 100 most upregulated genes were all upregulated >64 fold and included multiple genes for cytokines (IL-1β, IL-6, IL-8, IL-12, IL-27, IL-36 and oncostatin M) and chemokines (CXCL-1,-3,-5 and -6, CCL-1, -8, -15, -20 and CCR-5). We have previously described M.tb inducing a matrix degrading phenotype by upregulating numerous MMPs that degrade ECM without significant elevation of their specific inhibitors, the Tissue Inhibitors of Matrix Metalloproteinases (TIMPs1-4) [37]. Confirming this, genes for MMP-1 and -3 were amongst

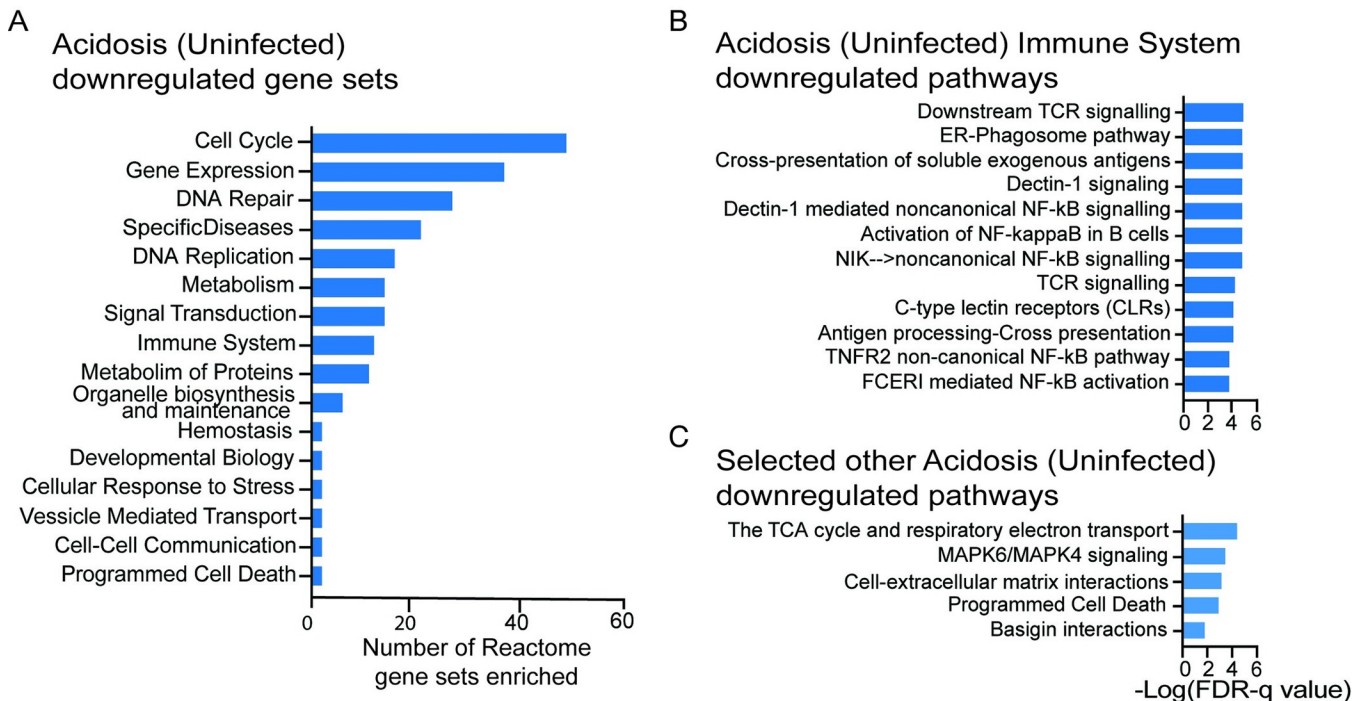

**Fig 2. Functional analysis of acidosis regulated genes in uninfected macrophages.** Gene Set Enrichment Analysis (GSEA) was performed on gene lists in the Acidosis (Uninfected) dataset ranked by fold change or significance (*p* value). Gene sets were considered enriched if FDR *q*-value is <0.05 in both fold change and significance ranking. Plotted FDR-q value is significance ranking value. No gene sets were enriched amongst upregulated genes. (A) The 219 Reactome gene sets downregulated by acidosis (pH 7.0) in uninfected cells. Gene sets are categorised by the uppermost level of the Reactome pathway event hierarchy. (B) Specific Immune System gene sets downregulated by acidosis. (C) Selected acidosis downregulated gene sets of importance to macrophage M.tb responses.

the top 100 most upregulated genes and multiple other MMPs including MMP-2, -7, -10, -12 and -14 were also upregulated. The key collagenase MMP-1 was upregulated 95.5 fold and its activators MMP-3 and MMP-10 upregulated 249.2 and 49.7 fold respectively (all p<0.01). Unlike the MMPs, of the TIMPs only TIMP-1 was upregulated > 2 fold (3.6 fold, p = <0.007) (Volcano plots of MMP expression in S2A Fig). MMPs are zinc dependent endopeptidases and members of the Metallothionein (MT) family, a group of zinc chaperone proteins that regulate intracellular zinc availability, were also amongst the highest upregulated genes. MT1 genes mt1a, mt1m, mt1h, mt1g, mt1jp were all upregulated >100 fold (all p<0.001) and mt1a was the third most upregulated gene by M.tb at pH 7.4 (642 fold).

GSEA showed enrichment of Immune System and Signal Transduction genes, including Type I and II interferon signalling, cytokine and chemokine pathways and NF-κB activation pathways (S2B Fig). Downregulated gene sets were mainly involved in Metabolism of Proteins, Cell Cycle and Metabolism. M.tb infection at pH 7.4 downregulated TCA cycle and Respiratory electron transport chain genes supporting a metabolic shift in infected MDMs towards aerobic glycolysis which is both important for intracellular killing of M.tb [38] and generates the lactic acid responsible for acidification of the inflammatory microenvironment.

## Extracellular Acidosis is proinflammatory in M.tb infected macrophages and drives Tissue Destruction pathways

M.tb infection at pH 7.0 (Infection pH 7.0) highly upregulated numerous genes for cytokines, chemokines and specific MMPs as seen with infection at pH 7.4. Similarly, many Immune System and Signal Transduction gene sets are enriched by M.tb infection at pH 7.4 and pH 7.0,

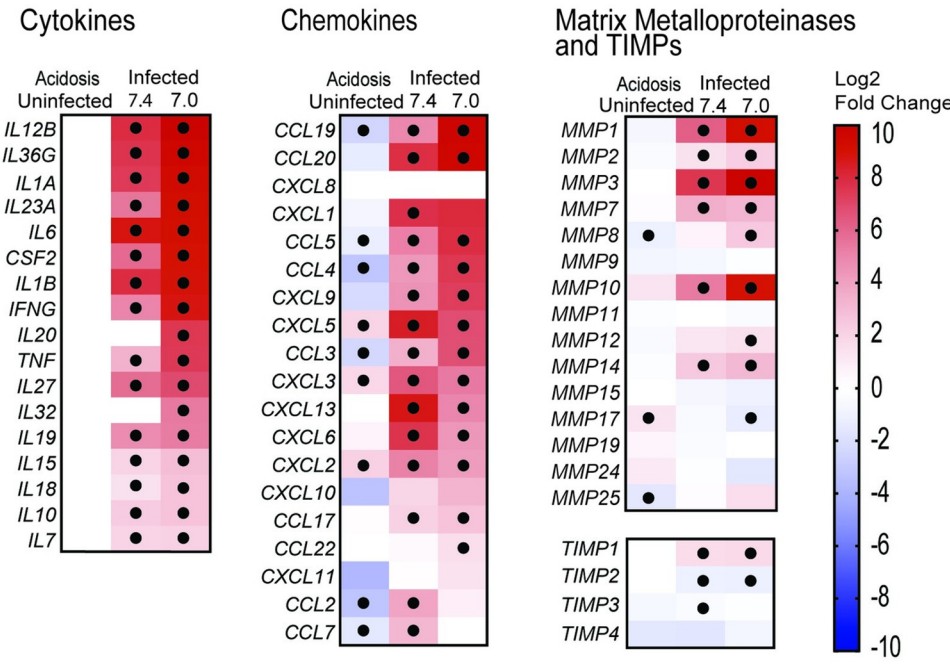

**Fig 3. Acidosis upregulates proinflammatory genes in M.tb infected macrophages.** Gene expression heatmaps of selected Cytokine, Chemokine, MMP and TIMP genes. Box shading indicates the the Log2 Fold Change in the Acidosis (uninfected), Infected (pH 7.4) and Infected (pH 7.0) analyses. Boxes unshaded (white) if gene expression not above threshold of detection. Black Circle indicates genes with individual $p$ value <0.01.

including genes for type 1 and 2 interferon responses, cytokine signalling, NF-κB activation and MAP kinases and PI3 kinase signalling. These are core transcriptional responses to M.tb infection independent of pH (S3A and S3B Fig).

Whilst in controls cells extracellular acidosis downregulated immune pathways, acidosis increased expression of many pro-inflammatory genes in M.tb-infected MDMs. Direct comparison of M.tb infected MDMs at pH 7.0 with pH 7.4 (Acidosis (Infected)) demonstrates that acidosis enhances expression of MMPs key to TB immunopathology including MMP-1, MMP-3 and MMP-10 which were increased 5.6, 4.3 and 7 fold respectively ($p<0.05$). Genes for proinflammatory TNF-α, IL-1β and IFN-γ, critical activators of antimycobacterial processes in macrophages, and IL-12, a key inducer of protective Th-1 responses [4], were also further upregulated >1.5 fold ($p<0.05$) by acidosis as were multiple chemokine genes (Fig 3).

To assess the overall effect of acidosis on M.tb infected macrophages we performed functional analysis on the results of the Acidosis (Infected) analysis. This demonstrated 22 upregulated Reactome reactions and 9 upregulated GO terms (Fig 4A and 4B). Gene sets relating to ECM organisation were the most frequently and significantly upregulated. Reactome gene sets ECM proteoglycans, Collagen Degradation, Integrin cell surface interactions, Degradation of the extracellular matrix and Extracellular Matrix Organisation and were all highly significantly upregulated (p<0.001, FDR-q<0.05). Extracellular Matrix Organisation is one of only 4 GO biological processes enriched along with Regulation of glucose metabolism, adrenal gland development and cell communication (all p<0.01, FDR-q <0.05). Also enriched were GO cellular functions Proteinaceous Extracellular Matrix and GO molecular function term Extracellular Matrix Structural Constituent. A total of 168 Reactome gene sets were downregulated in the Acidosis (Infected) analysis with the majority involved in the cell cycle, DNA repair and replication, specific disease states and metabolism of proteins similar to that seen in MDMs

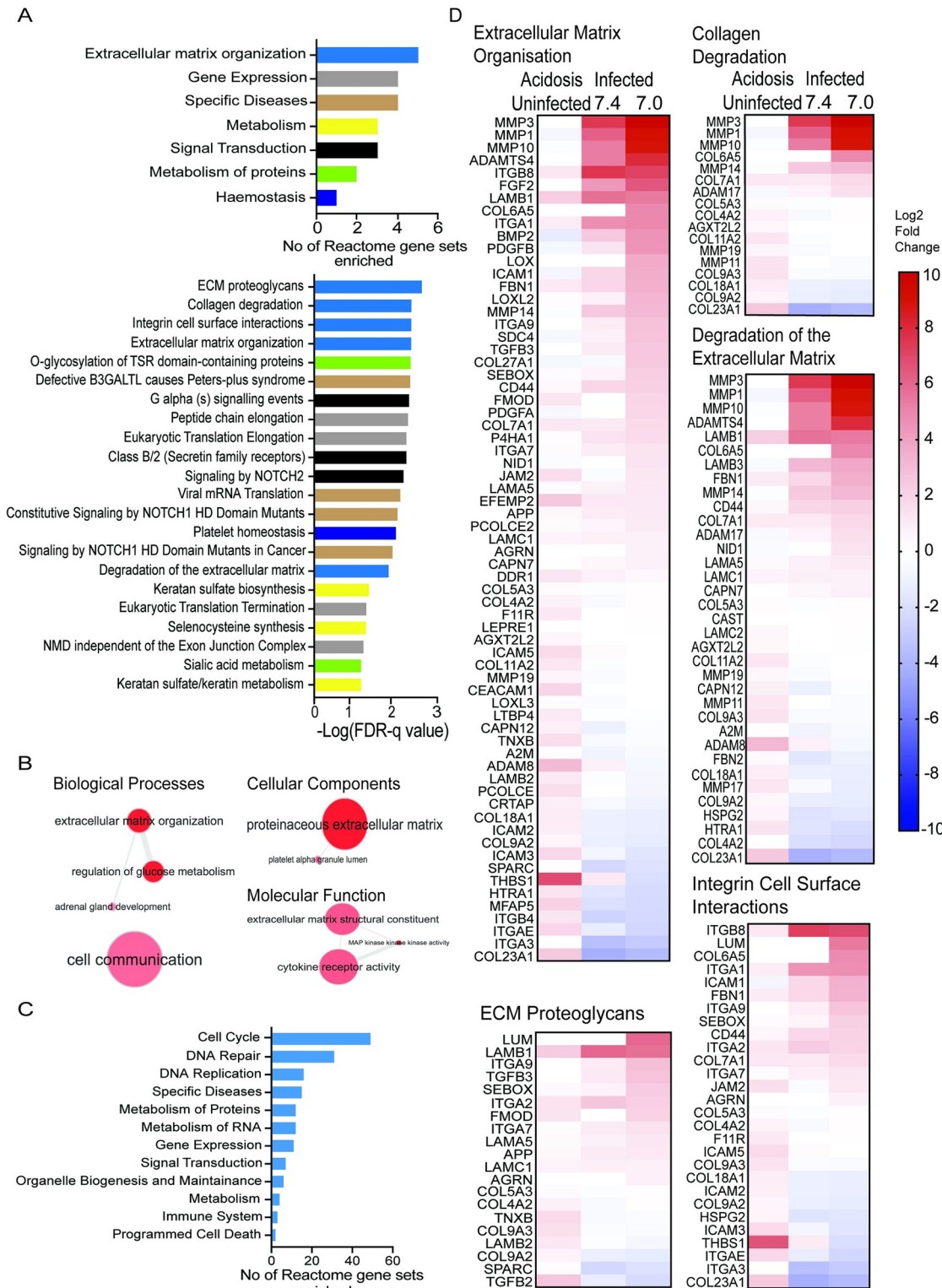

**Fig 4. Extracellular Matrix degradation pathways are enhanced by acidosis in M.tb infection.** GSEA of gene lists comparing M. tb infection of MDMs at pH 7.0 versus pH 7.4 (Acidosis (Infected)) shows upregulation of ECM degradation pathways due to increased MMP expression. (A) Broad classification of the 22 enriched Reactome pathways in the Acidosis (Infected) analysis (top panel) and the individual gene sets ordered by FDR q-value (bottom panel). (B) GO terms upregulated in Acidosis (Infected) analysis plotted using REViGO. Magnitude of bubble red shading indicates degree of significance and edges link related terms. (C)

Broad classification of the 168 Reactome downregulated gene sets. (D) Heatmap of gene expression of the individual genes included in the 5 upregulated Extracellular Matrix Organisation gene sets. Gene Log2 fold changes in the Acidosis (Uninfected), Infection (pH7.4) and Infection (pH 7.0) analysis (left to right) plotted. Only genes in the gene set with expression in one or more conditions are shown.

with acidosis alone. Only 3 of these downregulated gene sets were immune system sets; Cross presentation of soluble exogenous antigens, NF-κB inducing kinase non canonical NF-κB signalling and antiviral mechanism by IFN-stimulated genes (Fig 4C).

Examining individual genes included in the enriched Extracellular Matrix Organisation gene set shows that genes for MMPs-1,-3 and -10 were the highest ranked genes in the set and the genes most upregulated by M.tb infection with acidosis (Fig 4D). MMPs-1, -3 and 10 were similarly the most significantly acidosis upregulated genes amongst Collagen Degradation and ECM degradation genes. After MMP genes, the next most upregulated gene in these sets was protease ADAMTS4 which also degrades ECM components and is highly expressed at sites of TB infection [39]. Transcriptome profiling therefore demonstrates that acidosis enhances expression of ECM degradation pathways in M.tb infected MDMs by increasing expression of key MMPs of known importance to TB immunopathology.

### Extracellular Acidosis increases secretion of MMP-1 and -3 and IL-1β but suppresses TNF-α

As the transcriptome data showed acidosis enhancement of ECM degradation pathways due to upregulation of MMP genes, we next investigated changes in protein secretion to confirm phenotypic change. Secretion of MMPs and TIMPs broadly followed the transcriptomic data. In M.tb infected MDMs, acidosis increased secretion of MMP-1, 2.3 fold (p<0.001), and its activator MMP-3 [40] 4.9 fold (p<0.0001) but there was no change in MMP-10 concentrations (Fig 5A–5C). Baseline secretion of MMPs-1, -3 and -10 was unchanged by acidosis. Increases in MMP-1 gene expression were confirmed by qPCR which showed 2.8 fold increase with acidosis (p<0.001) (Fig 5D). No significant changes in secretion of TIMP-1 or -2 was detected, despite the observed modest increase in gene expression, confirming that acidosis worsens MMP/TIMP imbalance (Fig 5E and 5F). We next measured secretion of cytokines key to TB immunity. Acidosis in uninfected cells did not induce secretion of any measured cytokines (IL-1β, TNF-α, IFN-γ, IL-6, IL-8, IL-10). Secretion of IL-1β, an important macrophage activator in TB, was increased by acidosis (Fig 5G). Contrasting the transcriptome data however, key regulators of antimycobacterial responses TNF-α, IFN-γ and IL-6 were markedly suppressed by acidosis (Fig 5H and 5I). Secretion of IL-8 and anti-inflammatory IL-10 was unchanged (Fig 5K and 5L). Acidosis effects on TNF-α and MMP-1 were unchanged if macrophages were cultured in AB human serum rather than FCS (S4).

### Murine macrophages predominantly express TDAG-8 which regulates MMP and TNF-α secretion by M.tb infected human macrophages

To investigate mechanisms regulating enhancement of MMP secretion, we next determined if acidosis sensing GPCRs were expressed at sites of M.tb infection in a murine model. Immuno-histochemistry was performed on lung sections of mice infected with live virulent M.tb (Fig 6A). Areas of granulomatous inflammation stained brightly for TDAG-8 particularly the infiltrating inflammatory cells (Fig 6Aii). Staining for OGR-1 in inflamed areas of tissue was less marked but still evident compared to isotype control (Fig 6Aiii). Similarly, in primary human MDMs, TDAG-8 was the most abundantly expressed acidosis sensing GPCR by qPCR with lower and variable levels of OGR-1 expression. GPR4 expression was not detected

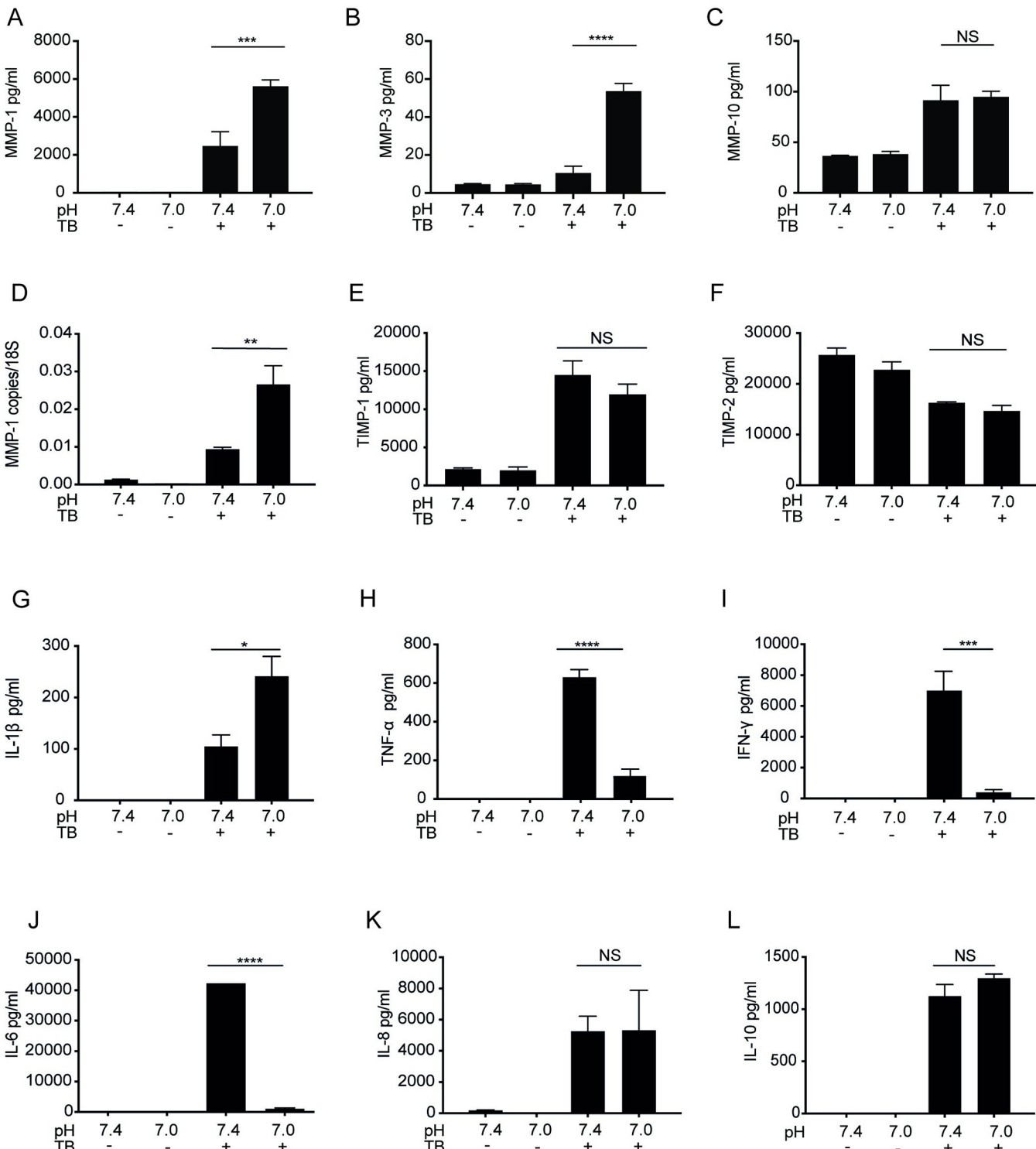

**Fig 5. Effect of Acidosis on secretion of MMPs, TIMPs and Cytokines from M.tb infected MDMs.** MDMs were infected with M.tb at pH 7.4 or pH 7.0 for 24 hours and supernatant protein levels measured at 24 hours. (A) MMP-1.(B) MMP-3. (C) MMP-10. (D) MMP-1 gene expression. (E) TIMP-1. (F) TIMP-2. (G) IL-1β. (H) TNF-α. (I) IFN-γ. (J) IL-6. (K) IL-8. (L). IL-10. Graphs are representative of n = 4 donors. Bars are means +/- SD. *p<0.05, **p<0.01, ***p<0.001, ****p<0.0001. NS–not significant.

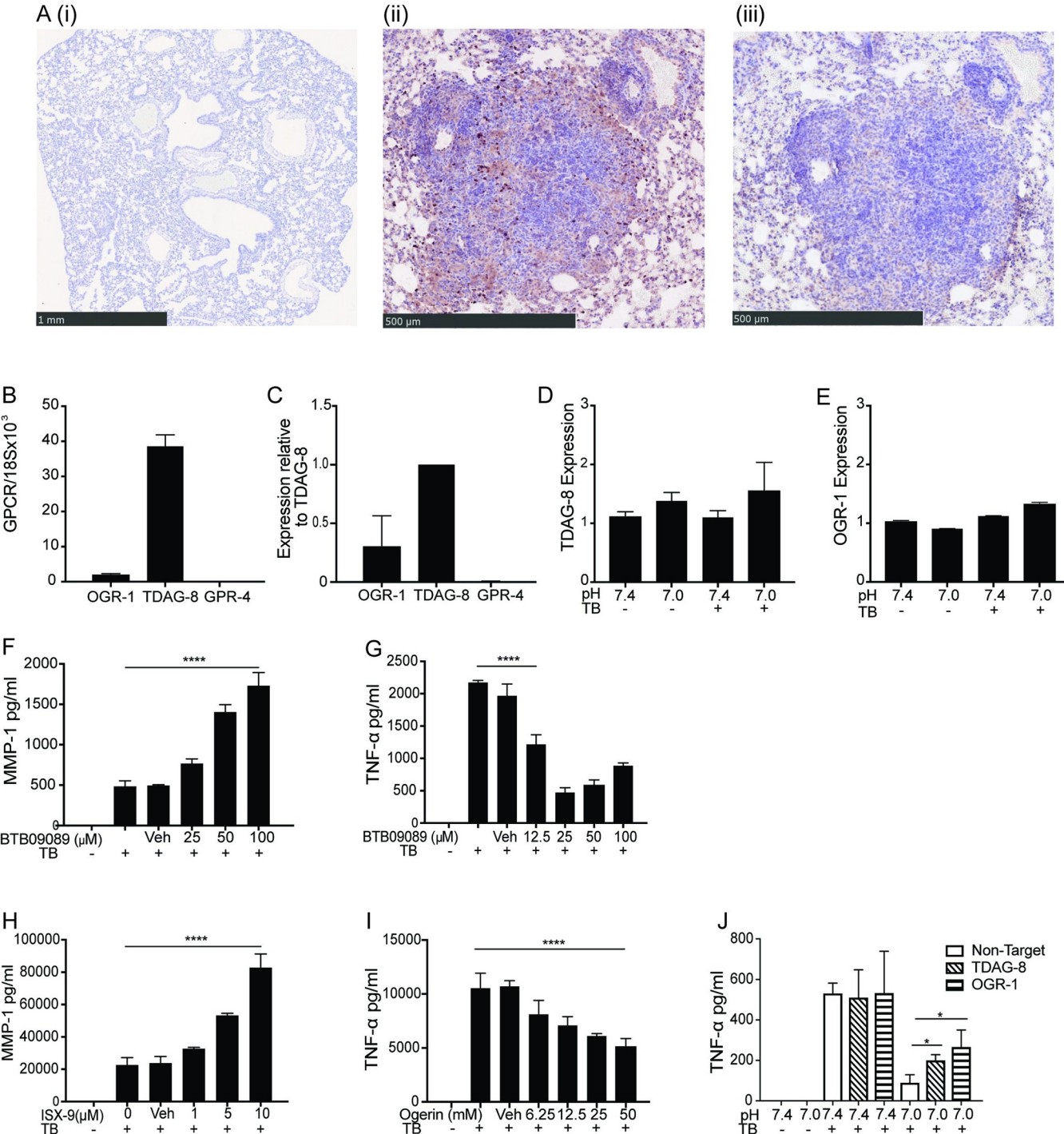

**Fig 6. Expression of TDAG-8 and OGR-1 and role of receptor signaling in mediating MDM acidosis responses.** (A) Lung sections of mice infected with M. tb by the intranasal route were stained for either (i) isotype control antibody (ii) TDAG-8 or (iii) OGR-1. Images representative from n = 3 animals. Gene expression of acidosis GPRCRs in primary human MDMs from healthy donors (n = 5) was measured by qPCR. (B) Representative relative expression of acidosis GPCRs from one human donor normalised to 18S. (C) Average expression across n = 5 donors normalised to TDAG-8 expression. Changes in expression of (D) TDAG-8 and (E) OGR-1 with M.Tb and acidosis. Effect of TDAG-8 agonist BTB09089 on (F) MMP-1 and (G) TNF-α secretion. Effect of OGR-1 agonists (E) ISX-9 on MMP-1 secretion and (F) Ogerin on TNF-α secretion. (J) Effect of gene knockdown of acidosis GPCRs on TNF-α secretion. B-J representative of n = 3 donors.

(Fig 6B and 6C). Expression of TDAG-8 and OGR-1 was unchanged by acidosis or M.tb infection (Fig 6D and 6E). To ascertain if the effect of acidosis was mediated by receptor signalling, agonist drugs and receptor RNA inference experiments were performed. TDAG-8 specific agonist BTB09089 [41] reproduces the effect of acidosis, increasing MMP-1 secretion 3.5 fold (p<0.0001) and suppressing TNF-α secretion 79% (p<0.0001) in a dose dependent manner (Fig 6F and 6G). The OGR-1 agonist ISX-9 [42] also increased MMP-1 secretion 3.7 fold (p<0.0001) whilst Ogerin, a partial allosteric modulator of OGR-1 [43], suppressed TNF-α secretion 51% (p<0.0001) (Fig 6H and 6I). Gene knockdown of both TDAG-8 and OGR-1 with siRNA, partially abrogates the acidosis induced suppression in TNF-α secretion demonstrating the importance of receptor signalling in mediating this response (Fig 6J).

## TDAG-8 is highly expressed in the lymph nodes of TB patients

To investigate the role of TDAG-8 in human immune responses to tuberculosis, we examined TDAG-8 expression in granuloma macrophages in patients with TB. Immunohistochemistry was performed on lymph node biopsies from 5 patients with culture proven TB lymphadenitis. All biopsies demonstrated necrotising granulomatous inflammation with multinucleate giant cell (MGC) formation characteristic of TB (Fig 7A). Compared to isotype control (Fig 7B),

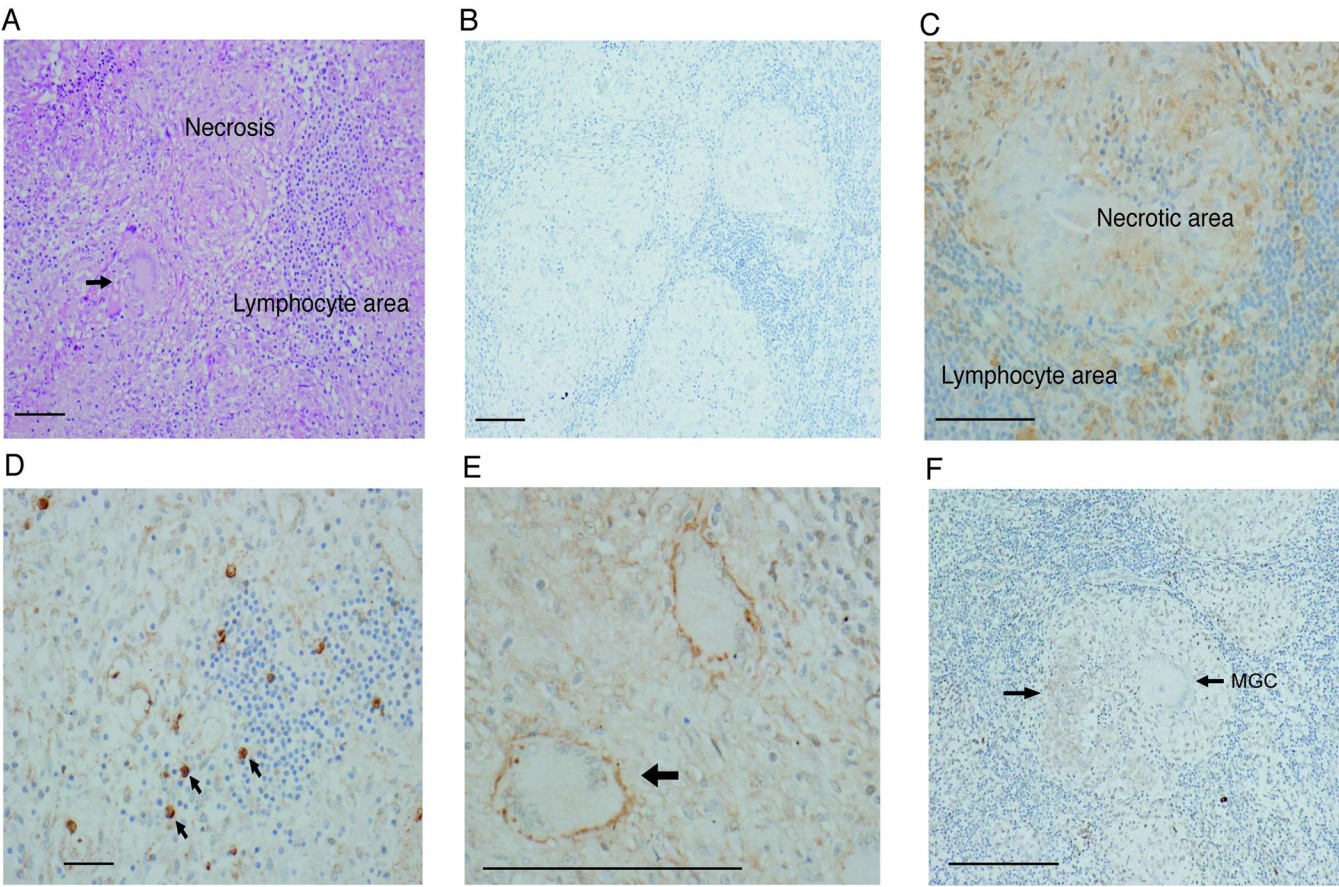

**Fig 7. TDAG-8 and OGR-1 expression in patients with TB Lymphadenitis.** Immunohistochemistry of lymph node sections from patients with TB lymphadenitis (n = 5). (A) Haemotoxylin/Eosin stain showing necrotising granulomatous inflammation with multinucleate giant cell (arrowed) characteristic of TB. (B) Isotype control staining. (C) TDAG-8 staining with increased staining in the macrophage containing granuloma center. (D) Infiltrating monocytes staining strongly for TDAG-8 (arrowed). (E) Multinucleate Giant Cells stain strongly for TDAG-8. (F) Staining for OGR-1. Staining in the center of the granuloma is marked with an arrow. Multinucleate Giant cell marked MGC in the figure. Images are representative of n = 5 donors. Scale bars 100µM.

TDAG-8 stained throughout the lymph node but staining was more marked in the necrotic centre of granulomas where M.tb-infected macrophages reside (Fig 7C). Cell specific staining was seen in infiltrating monocytes which stained brightly for TDAG-8 (Fig 7D) as well as Langhans type MGCs (Fig 7E). OGR-1 staining was present in the granuloma centre associated with necrosis (Arrowed in Fig 7F) but much less marked than for TDAG-8 and not apparently cell associated. Multinucleate Giant Cells did not demonstrate OGR-1 staining (Fig 7F).

## Extracellular lactate increases MMP-1 and TNF-α secretion by macrophages dependent upon Monocarboxylate transporter (MCT)-1

The microenvironment in inflammation is acidic specifically due to the build up of lactic acid and consequently the microenvironment in tuberculosis has high lactate concentration in addition to acidosis [44–46]. We therefore sought to determine if lactate has independent effects to acidosis on the pro-inflammatory response. Sodium L-Lactate (NaL) was used to manipulate media lactate concentration without altering pH. Lactate alone did not cause TNF-α or MMP-1 secretion by uninfected MDMs. Lactate increases TNF-α secretion from M.tb-infected MDMs 1.9 fold at 50mM (p<0.0001) but no effect was observed at lower concentrations (5-30mM NaL) (Fig 8A). MMP-1 secretion increased in a dose dependent fashion in response to NaL; 50mM NaL resulted in a 5.6 fold increase in MMP-1 concentrations (p<0.0001) (Fig 8B) and 25mM lactate increased MMP-1 gene expression 2.3 fold (Fig 8C). Neither acidosis or lactate had any effect on intracellular M.tb survival (Fig 8D). Finally, we investigated whether the effect of lactate required intracellular transport via MCTs which are known to be expressed in mouse lungs following M.tb infection [47]. Human MDMs expressed both MCT-1, a pH dependent importer of lactate [48] and MCT-4 a lactate exporter [48] (Fig 8E–8G). RNA-seq showed infection at pH 7.4 decreased expression of *SLC16A1* (MCT-1) 1.5 fold (p = 0.02) and increased *SLC16A3* (MCT-4) 1.6 fold (p = 0.004). We confirmed this with qPCR and further showed no expression change with lactate alone. Combined infection with lactate increased SLC16A3 (MCT-4) expression 2.5 fold (p<0.01) (Fig 8E and 8F) however this change was not reflected in western blots (Fig 8G). MCT-1 activity is critical to lactate regulation of MMP-1 secretion as blockade of MCT-1 with specific inhibitor AR-C122982 abrogated the effect of lactate showing that lactate acts intracellularly to increase MMP-1 secretion (Fig 8H).

## Discussion

The extracellular inflammatory microenvironment in TB is acidic due to lactic acid accumulation as a consequence of cellular metabolism. Using an *in vitro* model, we found that this extracellular acidosis generates a unique transcriptional signature in both M.tb infected and uninfected macrophages. In uninfected macrophages, modest transcriptional change was induced by acidosis, increasing expression of 117 genes and decreasing 23 genes. This change is broadly inhibitory with downregulation of genes involved in the cell cycle, gene expression, DNA repair and replication, signal transduction and metabolism. Similar transcriptional responses are seen in model organisms such as yeast, HeLa cells and human lung fibroblasts in response to numerous cellular stressors suggesting that acidosis activates the conserved "Environmental Stress Response" [33]. Acidosis stimulates IL-1β secretion in unstimulated human monocytes [49] however we found no macrophage cytokine gene expression or secretion was induced by acidosis alone. Acidosis has been proposed as a "novel danger signal" [26] but our data does not support this. No immune system gene sets were upregulated by acidosis alone, whilst antigen processing, Dectin-1 and NF-κβ signalling gene sets were downregulated. This

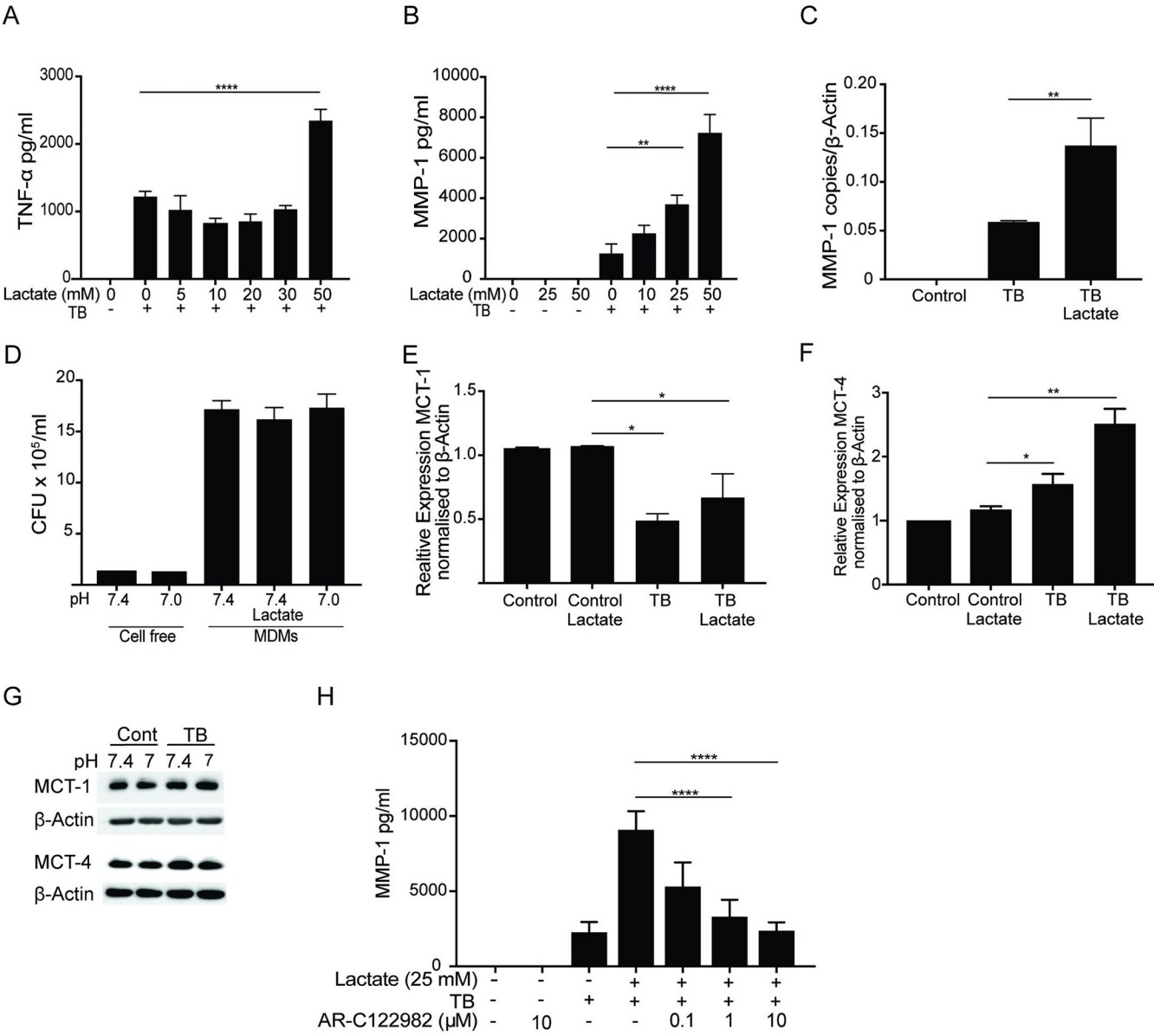

**Fig 8. Effect of Lactate on Macrophage MMP and Cytokine Secretion.** The effect of extracellular lactate at pH 7.4 on MDM secretion of (A) TNF-α and (B) MMP-1. (C) Gene expression of MMP-1with lactate (25mM). (D) Intracellular and cell free M.tb growth is unaffected by extracellular pH or lactate (25mM) at 24 hours. Gene expression of (E) *SLC16A1* (MCT-1) and (F) *SLC16A3* (MCT-4) with lactate and M.tb infection. (G) Western blots for MDM MCT-1 and -4. (H) Effect of an MCT-1 inhibitor on MMP-1 secretion. All experiments representative of n = 3 donors and performed with MDMs incubated for 24 hours. Bars are means +/- SD. *p<0.05, **p<0.01, ***p<0.001, ****p<0.0001.

contrasts with other well characterised endogenous "danger" signals such as hypoxia [50] and ROS [51] which increase cytokine and NF-κβ expression even in unstimulated cells.

M.tb infection, irrespective of pH, upregulated multiple cytokine and chemokine genes, cell surface activation markers, Interferon stimulated genes and glucose metabolism genes supporting the findings of others that these are core macrophage transcriptional responses to M. tb infection [52, 53]. Activated macrophages undergo "glycolytic reprogramming"; rapidly metabolising glucose via aerobic glycolysis yielding excess pyruvate which is converted to lactic acid by lactate dehydrogenase (LDH). Simultaneously, the TCA cycle and oxidative

phosphorylation is suppressed [54]. This increase in aerobic glycolysis produces the lactic acidosis which acidifies the inflammatory microenvironment. Supporting a metabolic shift in M. tb infected human MDMs we found increased expression of multiple glucose uptake (SLC2A1, SLC2A3, SLC2A6), glycolysis (HK2, PFKFB3, ALDOC) and Lactate Dehydrogenase (LDHA, LDHB) genes whilst the "TCA cycle and respiratory electron transport" was the 5th most significantly downregulated gene set by infection. Amongst genes regulated by infection with acidosis but not at pH 7.4 the Reactome gene set "glycolysis" was enriched and expression of glycolysis genes (HK2, ALDOC, LDHB) was increased by acidosis. GO biological process term "regulation of glucose metabolism" was one of only 4 GO processes upregulated in the Acidosis (Infected) analysis. This suggests acidosis increases glycolysis potentially favouring further acidification of the microenvironment. Metabolic reprogramming requires MCT-4 mediated lactate export [55], and MCT-4 expression is increased in granulomas of TB patients [56]. Expression of MCT-4 in MDMs was increased by M.tb however significant changes in protein expression were not observed, possibly as baseline MCT expression in MDMs was already high.

In contrast to uninfected macrophages, extracellular acidosis acts in marked synergy with M.tb to cause profound proinflammatory transcriptional change in infected MDMs. Comparing macrophages infected at pH 7.0, as encountered in the inflammatory microenvironment, with infection at pH 7.4 revealed specific upregulation of gene sets controlling ECM Organisation, Collagen Degradation and Degradation of the Extracellular Matrix. Within these gene sets, genes for MMPs-1, -3 and -10 were the most upregulated by acidosis, and we confirmed secretion of MMP-1 and its' activator MMP-3 was increased with acidosis. An acidic microenvironment activates MMP-3 [57] and MMP-3 catalytic activity is greater at low pH [58]. Extracellular lactate above 10mM; which occurs in meningeal [45], pleural [46] and peritoneal TB [17], also increased MMP-1 secretion in the presence of infection. Lactate effects on MMP-1 secretion was abrogated by inhibition of lactate transporter MCT-1 suggesting that lactate acts intracellularly to modulate MMP-1 secretion. Additionally acidosis increased IL-1β gene expression and secretion by M.tb infected macrophages and IL-1β drives lung parenchyma MMP production in TB [59]. Extracellular acidosis and lactate thus independently enhance secretion of MMP-1 and its activators from M.tb infected macrophages without significant change in inhibiting TIMPs. MMP-1 is increasingly recognised as the key driver of tissue destruction in TB. MMP-1 is one of few enzymes capable of degrading fibrillary collagen [60], the principal structural component of the lung. Macrophage MMP-1 secretion is required for collagen degradation [9] and caseating necrosis [10] in TB and MMP-1 is highly expressed in TB cavities [61]. In patients, sputum MMP-1 and -3 levels correlate with disease severity [9]. Microenvironment pH and lactate enhancement of macrophage MMP-1 activity may therefore worsen morbidity in TB patients however this requires confirmation in *in vivo* models.

TNF-α is critical for macrophage M.tb killing and granuloma formation [62], and patients receiving anti-TNF-α therapy are at high risk of TB reactivation [63]. Acidosis markedly suppressed TNF-α secretion by infected MDMs despite increases in TNF expression. This is reported in LPS stimulated macrophages due to both cytosolic retention [64] and post transcriptional regulation [65] of TNF-α. Acidosis reduces protein synthesis [66], and we found translation and protein synthesis gene sets were significantly reduced by acidosis in MDMs. Increases in MMP and IL-1β secretion with acidosis however demonstrates that TNF-α suppression is a specific effect rather than a consequence of globally reduced protein synthesis. Microenvironment lactate is unlikely to balance acidosis effects on TNF-α as lactate only increased TNF-α secretion at concentrations greater than 30mM, above that described in TB [44–46]. Acidosis additionally suppressed M.tb infected MDM secretion of IFN-γ whilst secretion of IL-10 was unchanged. Low TNF-α, INF-γ levels favours M.tb survival [3]. Despite this,

early intracellular M.tb infection burden, was unaffected by acidosis possibly as increased glycolytic rate and IL-1β stimulation increase macrophage intracellular killing [67]. This observation also suggests uptake of M.tb by MDMs was not significantly changed by acidosis.

The mechanism by which acidosis might affect macrophage function and gene expression in acidosis was unknown so we investigated the OGR-1 family of GPCRs comprising OGR-1, TDAG-8, GPR4 which signal extracellular acidosis within the extracellular pH range in TB [68]. In a murine model of TB, we found inflammatory lesions stained clearly for TDAG-8 and OGR-1 with numerous inflammatory cells staining brightly for TDAG-8. We confirmed human MDMs express both TDAG-8 and OGR-1, consistent with previous data [24], but TDAG-8 expression was significantly higher. In human monocytic cell lines acidosis, hypoxia and LPS all increase acidosis receptor expression [69] but expression in primary human MDMs was unchanged by acidosis or M.tb infection. Granulomas from TB patients stained strongly for TDAG-8, particularly MNGs which are of the macrophage lineage and a histological hallmark of TB disease. TDAG-8 agonists had similar effects on MMP-1 and TNF-α concentrations to those observed in acidosis whilst receptor knockdown abrogates acidosis TNF-α suppression. OGR-1 knock down also reduced acidosis TNF-α suppression. OGR-1 expression is lower than TDAG-8 expression in MDMs, and expression at sites of infection in patients was less clear. Lung OGR-1 expression is predominantly by epithelial and smooth muscle cells and whole lung expression is lower than for TDAG-8 [23]. Together these observations show that microenvironment acidosis, signalled via TDAG-8 expression in M.tb infected macrophages suppresses inflammatory, and enhances ECM turnover pathways. This effect on macrophages may explain the observations that TDAG-8 knock out mice have exaggerated delayed type hypersensitivity and reduced pathology in a collagen induced arthritis model [22]. These central findings are illustrated in Fig 9.

In summary, we show that extracellular acidosis and lactate in the inflammatory microenvironment modulates M.tb infected macrophage responses favouring MMP-1 activity and tissue degradation pathways which can be predicted to worsen disease pathology. Acidosis, but not lactate, suppresses TNF-α secretion from infected MDMs. Acidosis effects on macrophage MMP and cytokine secretion are dependent upon acidosis receptor signalling of which TDAG-8 is the most abundantly expressed. Fully elucidating how acidosis and TDAG-8 regulates inflammatory responses in the complex environment of TB lesions in patients requires further study and may identify a new target for host directed therapy.

## Methods

### Ethics statement

Patients were recruited from several hospital sites in Lima, Peru. Written informed consent was obtained from each patient and study protocol was approved by Local and National Research Ethics Committee in Peru. All animals were handled in accordance with UK national Home Office regulations and all procedures were approved locally by Imperial College, London.

### M. tuberculosis culture

*M. tuberculosis* H37Rv, obtained from The National Collection of Type Cultures (UKHSA, UK), was cultured in Middlebrook 7H9 media (BD Diagnostics) supplemented with 10% ADC enrichment medium (BD Diagnostics), 0.2% glycerol and 0.02% Tween 80. Clumping was prevented by circular agitation (100rpm) and brief sonication prior to use. M.tb was grown to mid log phase and used at a multiplicity of infection (MOI) of 1 as described previously [70].

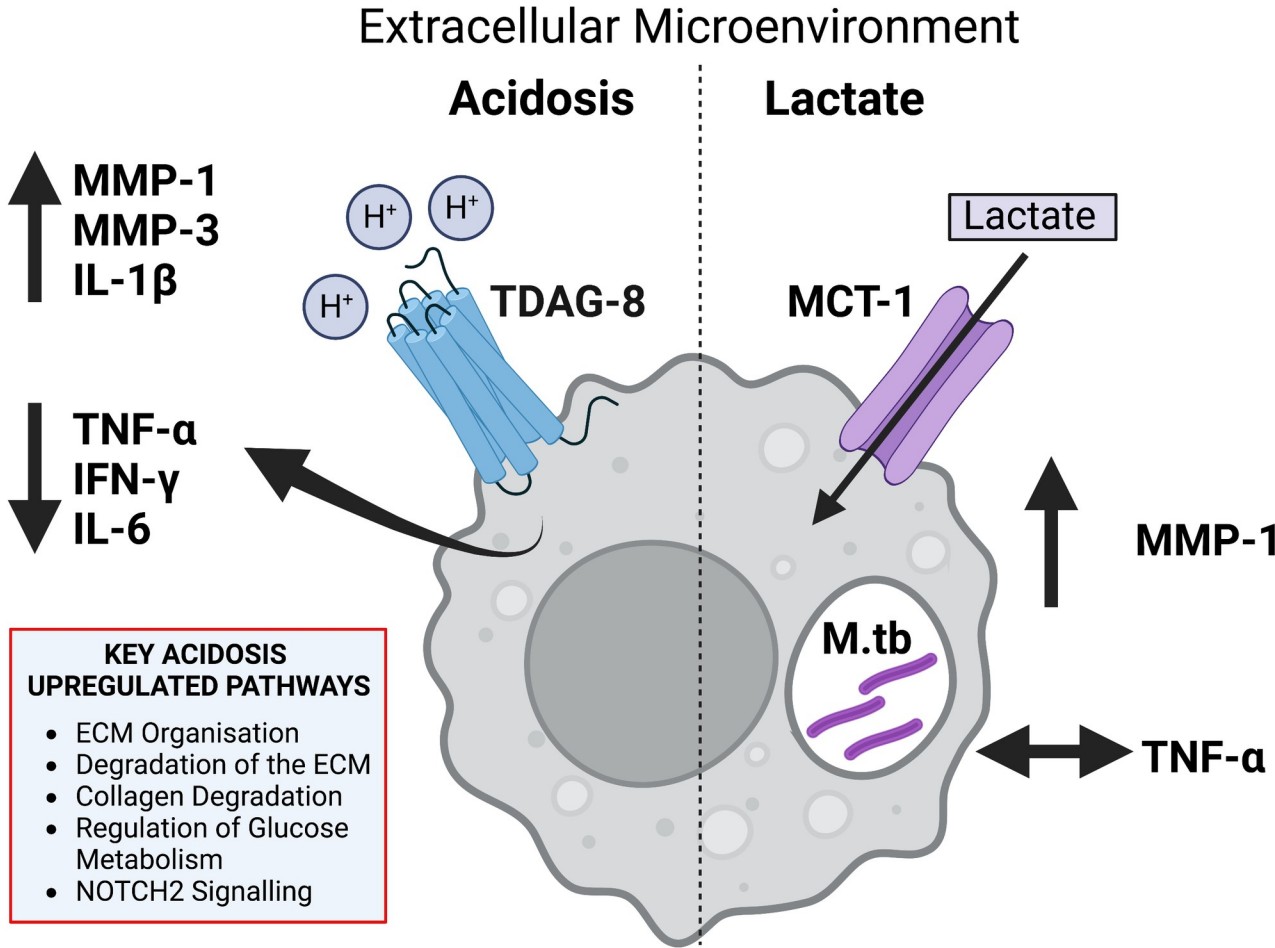

**Fig 9. Summary of the effects of microenvironment acidosis and lactate on M.tb infected MDMs.** Review of the main changes in protein secretion and gene transcription in M.tb infected MDMs exposed to extracellular acidosis and lactate.

## Macrophage and monocyte culture

Peripheral Blood Mononuclear Cells (PBMC) were isolated from single donor leukocyte cones purchased from the National Blood Transfusion Service (Colindale, UK) by density gradient centrifugation and adhesion purification. Briefly, leukocytes were diluted 1:2 with Hanks-Balanced Salt Solution (HBSS) (Gibco), layered on Ficoll-Paque (GE Healthcare) and centrifuged at 1500 rpm for 30 mins. The PBMC layer was aspirated, washed and monocytes were plated at 250,000 monocytes/cm$^2$ in RPMI 1640 supplemented with 2mM glutamine and 10μg/ml ampicillin (Gibco). After 1 hour, non-adherent cells were removed by washing with HBSS. Monocyte Derived Macrophages (MDMs) were generated by culturing monocytes for 4 days in RPMI 1640/2mM glutamine/10μg/ml ampicillin/10% heat inactivated Fetal Calf Serum (FCS) (Biowest) and 100ng/ml of Macrophage Colony Stimulating Factor (M-CSF) followed by 24 hour culture in RPMI/10%FCS without growth factors. Experiments were performed using Macrophage Serum Free Media (M-SFM) (Gibco). Purity of macrophages was >94%.

## Adjusting pH and Lactate concentration of cell culture media

The pH of cell culture media was adjusted with predetermined volumes of 1M HCL or 1M NaOH. Media was incubated at 37˚C and 5% $CO_2$ for 24 hours to reach equilibrium. pH was

measured with a 3 point calibrated pH meter accuracy +/- 0.01. 1M Sodium L-Lactate (Sigma-Aldrich) was used in lactate experiments and did not affect pH. Cell viability at 24 hrs was unaffected by the pH range used (pH6.8–7.4) and >95% in all experiments. Viability was assessed by measuring supernatant LDH activity using LDH Cytotoxicity Detection Kit (Roche) according to manufacturers instructions and exclusion of fluorescent DNA binding dye Fixable Viability Dye eFlour450 (eBiosciences) by flow cytometry using a BD FACSCalibur (BD Biosciences).

## RNA-seq

At 24 hours for each condition $2.5x10^6$ macrophages were lysed in Tri-Reagent RNA isolation buffer (Sigma-Aldrich) and purified by spin column purification with in column DNAase step using DirectZol Miniprep kit as per manufactures instructions (Zymo Research). Only samples yielding >2.2μg of RNA with RNA Integrity Number > 7.0 were used to generate a stranded, polyadenylated mRNA focused library using a Tru-seq Stranded mRNA Library Prep Kit (Illumina). Sequencing was performed on the Illumina HiSeq 4000 platform (Edinburgh Genomics) using a single flow cell lane to generate >30 million 2x75bp paired end reads per sample. Reads were aligned to a reference *Homo Sapiens* genome (Ensembl v 37.71) using STAR version 2.5 and the resulting BAM files sorted with Picard-tools (version 1.115) using default parameters. Read counts were generated using htseq-count version 0.6.0 and python version 2.7.3 with parameters "-m union–s reverse–I gene id–t exon" using ensemble gene annotations. Genes with less than 20 counts mapping across all samples were excluded from analysis. Differential expression analysis was performed using Bioconductor package EdgeR (version 3.10.5) with R version 3.2.1. The Quasi likelihood F-test approach was used due to the low replicate number. Genes with an individual False Discovery Rate (FDR) of <0.01 were considered differentially expressed. A fold change cut off was not used. Functional analysis was performed using Gene Subset Enrichment Analysis (GSEA) from the Broad Institute (version 2.2.1). GSEA was performed on a pre ranked list of all genes in each condition ranked either by fold change, or p value (p) using metric $-1*(\log(1/p))$. Displayed results show gene sets enriched in both fold change and significance analyses. Analysis parameters were: permutations 1000, Maximum gene set size 500, Minimum gene set size 15, weighted analysis. Gene sets from both the Gene Ontology and Reactome databases were used with similar results. GSEA results displayed in S2 Dataset shows Size (number of genes in the gene set after excluding genes not expressed in the dataset), Nominal p value (Statistical significance of the enrichment score), False Positive Rate q value (estimated probability that the normalized enrichment score represents a false positive finding), Rank at Max (position in the ranked list at which the maximum enrichment score occurs) and Leading Edge data. GO terms enriched in the analysis are displayed using REVIGO [71] where bubble shading indicates significance (darker more significant) and edges link related GO terms. The raw sequencing data has been deposited in the European Nucleotide Archive (ENA) under accession number PRJEB20229 and is publicly available.

## Intracellular *M. tuberculosis* survival

Infected MDMs were lysed in sterile water after 24 hours, serially diluted and plated onto 7H11 agar (Middlebrook). Individual colony forming units were counted after incubation at 37˚C for 2–3 weeks.

## Enzyme Linked Immunosorbent Assay (ELISA) and Luminex Muliplex Assay

Levels of MMPs, TIMPs and cytokines were measured by ELISA assay (R&D duoset, R&D systems) according to the manufacturer's instructions. Where stated in results, levels of MMPs

and cytokines in supernatants were also measured by multiplex microsphere based immunoassay on the Bio-Rad Bio-Plex 200 system (Bio-Rad Laboratories). For MMPs the Human Luminex Performance Assay base Kit (R&D systems) was used in conjunction with selected human MMP primary and secondary magnetic bead sets (R&D systems) according to manufacturer's instructions. Cytokine levels were measured using Human Cytokine Multiplex Assay (Millipore).

## Western blot

Samples were washed in phosphate buffered saline and lysed in SDS lysis buffer (2%SDS, Tris 62.5mM, glycerol 10%, DDT 50mM) heat denatured at 70˚C for 10mins and centrifuged at 10000rpm for 1min before loading onto a NuPAGE 4–12% Bis Tris Gel (Life Technologies). Proteins were separated by electrophoresis using the SureLock MiniCell system (Life Technologies) and MES running buffer (Life technologies) then transferred to nitrocellulose membrane (GE Healthcare) using an X-Cell Blot Module and Transfer buffer (Life Technologies). Membranes were blocked for 1 hour (5% non-fat dry milk, 0.1% Tween-20 in Tris buffered saline) and incubated overnight with primary antibody diluted in blocking buffer at 4˚C. Membranes were then incubated with secondary HRP-conjugated antibody, diluted in blocking buffer for 1 hour at room temperature. Blots were developed with Bio-Rad Clarity (Bio-Rad) for 1–2 mins and chemiluminescence imaged using the Chemidoc MP Imaging system (Bio-Rad, UK). Image Lab v5.2.1 (Bio-Rad, UK) was used for post acquisition image processing. Antibodies used MCT-1 (Santa Cruz, sc-365501), MCT-4 (Santa Cruz, sc-50329), β-Actin (Cell Signalling, 8H10D10).

## RNA Extraction, cDNA Synthesis and Real Time PCR

RNA was extracted as previously described. One microgram of RNA was then reverse transcribed to cDNA using the QuantiTect Reverse Transcription Kit (Qiagen). Resulting cDNA was then used in an RT-PCR reaction using specific primer and probe mixes with Brilliant II mastermix (Agilent) on the Stratagene Mx3000p platform (Agilent). The following thermal profile was used: 10min at 95˚C followed by 40 cycles at 95˚C for 30 sec then 60˚C for 1 min. For MMPs genes cycle thresholds were compared to a plasmid standard curve of known MMP and housekeeping gene concentrations. For other genes relative gene expression was calculated using the Pfaffl comparative method. The following custom oligonucleotides (MMP-1 forward primer 5'- AAGATGAAAGGTGGACCAACAATT -3'; Reverse primer 5' -CCAAGAGAAT GGCCGAGTTC -3'; Probe 5'- FAM CAGAGAGTACAACTTACATCGTGTTGCGGCTC-T AMRA -3', Sigma) and proprietary mixes were used (Thermofisher TaqMan reagents; OGR-1 Hs00268858_s1, TDAG-8 Hs00269247_s1, SCL16A1 Hs01560299_m1, SLC16 A3 Hs00358829_m1, 18S Hs03003631_g1, ACTB Hs01060665_g1)

## Macrophage Transfection with siRNA

MDMs were generated as previously described and seeded at 105,000 cells/cm$^2$. Transfection with DharmaFect 3 transfection reagent (Dharmacon) was performed as per manufactures instructions. For each gene a mix of 4 specific "On-target plus Smart-pool" siRNAs was compared with a similar pool of 4 non-targeting control siRNA (Dharmacon). Transfection was performed in antibiotic/serum free RPMI 1640 supplemented with 2mM glutamine in 12 well plates. 5μl/well of DharmaFect 3 was suspended in the appropriate concentration of serum free media. The siRNA was suspended in siRNA buffer (Dharmacon) and diluted in media to a final concentration of 25nM. The transfection reagent and siRNA were combined by gentle agitation at 20˚C for 20mins, cells transfected for 8 hours then rested for 12hrs in RPMI 1640

(2mM glutamine, 10μg/ml ampicillin, 10% heat FCS). Transfection efficiency as measured by uptake of florescence (FAM) labelled siRNA was > 95%. The described protocol was optimised for cell seeding density, transfection reagent concentration and transfection duration and yielded target gene expression compared to control of 30% for TDAG-8 and 49% for OGR-1.

### Immunohistochemistry of TB Tissue

Lymph node biopsies were taken from HIV negative, adult patients with culture proven TB lymphadenitis. All biopsies had the typical histological features of TB lymphadenitis as confirmed by 3 independent, blinded, histopathologists. Additionally, paraffin fixed lung tissue blocks were obtained from Balb/C mice infected with $10^4$ live virulent M. tuberculosis H37Rv by intranasal installation. Paraffin fixed patient biopsies were deparaffinised by heating (60˚C for 8mins) and staining performed on the Ventana Benchmark Ultra platform (Ventana Inc). Antigen retrieval was performed by incubation with Ventana Protease 3 cocktail (Ventana Inc). Primary antibody (1:300) was incubated for 32 mins at 32˚C. Endogenous peroxidase activity was blocked with Ventana Optiview Peroxidase Inhibitor (3%) and visualization performed with Ventana Optiview DAB Detection Kit and counterstained with Haematoxylin II/ Ventana Bluing Reagent (Ventana Inc). For animal studies cut sections of paraffin embedded mouse lung were deparaffinised and heat induced antigen retrieval performed. Primary antibody (1:50) was incubated at 4˚C overnight in blocking buffer (4% goat serum/1%BSA/PBS). Slides were washed with PBS and incubated for 45mins at room temperature with secondary antibody (1:200) in blocking buffer followed by detection using NovaRed HRP Detection Kit (Vector Labs). Antibodies used; Rabbit anti-human TDAG-8 IgG (ab188907, Abcam), Rabbit anti-human OGR-1 IgG (ab72500, Abcam), biotinylated anti-rabbit IgG1 (BA-1000, Vector labs)

### Reagents

MCT-1/2 inhibitor AR-C155858, TDAG-8 agonist BTB09089 and OGR-1 agonist ISX-9 were purchased from Tocris Bioscience and OGR-1 allosteric agonist Ogerin was purchased from Sigma. Where drugs were not dissolvable in cell culture media, DMSO was used at final concentration <1% and appropriate vehicle controls included.

### Statistical analysis

Statistical analysis of sequencing data was performed as described in detail in the RNA-seq methodology section. Statistics on cell culture experiments (Figs 4–8) was performed using Prism Graphpad Software v7. Unless otherwise stated in the text all experiments were performed in triplicate and displayed results are representative of a minimum of 3 independent experiments with separate donors. Significance was assessed by 1-way ANOVA with Tukey's test except for experiments with repeated conditions when a 2 way ANOVA with Bonferoni correction was used. Results were considered statistically significant if $p < 0.05$. Graphs display the mean +/- SD unless otherwise stated. On graphs statistical significance between conditions is marked with *. *$p < 0.05$, **$p < 0.01$, ***$p < 0.001$, ****$p < 0.0001$.

### Supporting information

**S1 Fig. PCA plot and Smear Plots showing average expression and fold change.** (A) Principal Component Analysis (PCA) showing the second and third principal component on the x and y axis respectively. Total variance attributable to that PC indicated. (B) Smear plots of gene expression in the Effect of Acidosis (uninfected), Effect of Infection at pH 7.4 and pH 7.0,

and the Effect of Acidosis (infected) analyses. $Log_2$ Fold Change on the y-axis and Log Counts per Million (CPM) x-axis. Individual genes with FDR<0.01 plotted in red.
(TIF)

**S2 Fig. Gene sets regulated by M.tb infection of MDMs at pH 7.4.** (A) Volcano plot with differential expression results by infection at pH 7.4. MMP genes labelled. Significantly regulated genes marked in red. (B) Reactome Gene sets enriched amongst genes upregulated by M.tb infection of MDMs at pH 7.4. Top panel shows number of enriched gene sets ordered by the uppermost level of the Reactome pathway event hierarchy. Gene sets were considered enriched only if FDR q-value <0.05 when genes were ranked by both fold change and significance (*p* value) to highlight the most significantly regulated gene sets. Lower panel displays individual gene sets. Plotted FDR-q value is significance ranking value. (C) Downregulated gene sets.
(TIF)

**S3 Fig. Gene Sets regulated by Infection at pH 7.4 and pH 7.0.** Results of GSEA when genes in the Infected (pH 7.4) and Infected (pH 7.0) analysis are ranked by *p* value (A) Comparison of the number of gene sets up and down regulated by Infection (pH 7.4) and Infection (pH 7.0) classified by the uppermost level of the Reactome pathway event hierarchy. (B) Gene sets enriched in both the Infection (pH 7.4) and the Infection (pH 7.0) analysis.
(TIF)

**S4 Fig. Macrophage MMP-1 and TNF-α secretion when cultured with AB Human Serum.** M.tb infected macrophages were cultured as described in the methods section except 10% AB Human serum was used rather than FCS. Graphs show MMP-1 expression (A) and secretion (B) and TNF-α secretion (C) after 24 hours. Graphs are representative of n = 2 donors. Bars are means +/- SD. *p<0.05, **p<0.01, ***p<0.001, ****p<0.0001. NS–not significant.
(TIF)

**S1 Dataset. File containing lists of differentially expressed genes in each analysis performed.**
(XLSX)

**S2 Dataset. File containing Gene Set Enrichment Analysis outputs for each analysis performed.**
(XLSX)

## Acknowledgments

This project was supported by the NIHR Clinical Research Facility at the Royal Brompton and Harefield NHS Foundation Trust and Imperial College London. Sequencing was performed by Edinburgh Genomics. Biorender.com was used to create some figures.

## Author Contributions

**Conceptualization:** Ashley M. Whittington, Brian D. Robertson, Joanna C. Porter, Robert H. Gilman, Jon S. Friedland.

**Formal analysis:** Ashley M. Whittington, Frances S. Turner, Candice Roufosse.

**Funding acquisition:** Ashley M. Whittington, Jon S. Friedland.

**Investigation:** Ashley M. Whittington, Frances S. Turner, Friedrich Baark, Sam Templeman, Daniela E. Kirwan, Candice Roufosse, Nitya Krishnan, Deborah L. W. Chong.

**Writing – original draft:** Ashley M. Whittington, Frances S. Turner, Friedrich Baark, Sam Templeman, Daniela E. Kirwan, Candice Roufosse, Nitya Krishnan, Brian D. Robertson, Deborah L. W. Chong, Joanna C. Porter, Robert H. Gilman, Jon S. Friedland.

**Writing – review & editing:** Ashley M. Whittington, Frances S. Turner, Friedrich Baark, Sam Templeman, Daniela E. Kirwan, Candice Roufosse, Nitya Krishnan, Brian D. Robertson, Deborah L. W. Chong, Joanna C. Porter, Robert H. Gilman, Jon S. Friedland.

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
