## [Decision Letter · Decision Letter 0]

25 Jan 2023

Dear Dr. Whittington,

Thank you very much for submitting your manuscript "An acidic microenvironment in Tuberculosis increases extracellular matrix degradation by regulating macrophage inflammatory responses" for consideration at PLOS Pathogens. As with all papers reviewed by the journal, your manuscript was reviewed by members of the editorial board and by several independent reviewers. In light of the reviews (below this email), we would like to invite the resubmission of a significantly-revised version that takes into account the reviewers' comments.  We would note that substantial concerns were raised both with regard to the interpretation of the disparate data sets, and likely the need for additional experiments to address the concerns of the reviewers.

We cannot make any decision about publication until we have seen the revised manuscript and your response to the reviewers' comments. Your revised manuscript is also likely to be sent to reviewers for further evaluation.

Sincerely,

David M. Lewinsohn

Academic Editor

PLOS Pathogens

Marcel Behr

Section Editor

PLOS Pathogens

Kasturi Haldar

Editor-in-Chief

PLOS Pathogens

orcid.org/0000-0001-5065-158X

Michael Malim

Editor-in-Chief

PLOS Pathogens

orcid.org/0000-0002-7699-2064

Reviewer's Responses to Questions

**Part I - Summary**

Reviewer #1: This manuscript explores an important topic, i.e., the impact of acidosis generated by tissue inflammation during M.tb infection. Using RNA-seq they show that extracellular acidosis and lactate in tissue culture modulates human macrophage responses in both unstimulated and especially M.tb-infected macrophages. They highlight effects on MMPs and tissue destruction and implicate acidosis sensing receptors, TDAG-8 and to some extent OGR-1, in the response (including immunohistochemistry expression data in mouse and human lungs, and agonist and siRNA KD data in macrophages) as well as MCTs, intracellular transporters. The model design used is interesting (changing pH and addition of lactate) and the results provocative. However, the various data presented are not clearly brought together into a cogent story with many unanswered questions. The discussion largely recapitulates the data and review of the literature but does not aid in an in depth understanding of the data presented. Thus, the significance of the work is in question. Including a model for the potential pathways involved would help. Also, the idea that TDAG-8 itself could be a host-directed therapy for TB patients is a stretch based on the data presented and the fact that blocking this receptor is expected to have many off target effects. Finally, some important experimental details are lacking. Additional comments follow.

Reviewer #2: With this manuscript, Whittington and colleagues have undertaken an elegant series of experiments to enhance our understanding of the factors that mediate lung damage during TB disease. They first used RNAseq to demonstrate that extracellular acidosis led to an upregulation of ECM degradative pathways in human macrophages, with increased expression of MMP-1 and -3. Acidosis also suppresses expression of key pro-inflammatory cytokines including TNF-a and IFN-g, indicating that acidosis alone essentially functions as an anti-inflammatory, including along pathways that are important for Mtb-specific responses. In contrast, and not surprisingly, Mtb infection (albeit at a pH of 7.4) led to upregulation of numerous genes and pathways associated with a classic pro-inflammatory, antimicrobial response and the accompanying shift in cell metabolism to aerobic glycolysis. This upregulation extended to numerous matrix-destroying MMPs (some of which had among the greatest fold differences in expression), without accompanying increases in TIMP levels. Many of these increases in pro-inflammatory, matrix-destroying transcriptional responses to Mtb and their accompanying pathways were further augmented in the presence of acidosis. Interestingly, protein secretion of MMPs mirrored the transcriptional changes, but levels of several key pro-inflammatory cytokines (IFNg, TNFa, IL6) were actually suppressed in Mtb-infected cells exposed to acidosis. They found that the effects of acidosis are mediated, at least in part, by acidosis-sensing GPCRs (TDAG8 > OGR1), and that lactate (and its transporter MCT-1) affected MMP-1 but not TNFa levels.

In summary, they have demonstrated a synergistic effect of Mtb infection and acidosis on macrophage function with potentiation of MMP expression and matrix-destroying pathways. These data are clinically relevant in that modulation of the local host environment pH, e.g., of TDAG8 activity, could be leveraged for host-directed therapies to reduce matrix destruction and improve long term lung outcomes among those with TB disease.

Reviewer #3: The paper by Whittington et al. examines a novel, interesting and important relationship between gene expression, acidosis, and Mtb infection. Overall the paper is clearly written and the data are displayed in an informative way. I have no major issues with the experiments or the validity of the results. I have two major points. Given the current dataset, only one of these, the latter, can be addressed without additional animal model experiments. The latter however could also be addressed with simply editing text.

1) Limitations of the models. Extracellular acidosis clearly appears to act in a synergistic way with Mtb infection to cause proinflammatory gene expression in human MDMs. Furthermore, in infected MDMs, acidosis specifically was associated with increased expression of MMP1, MMP3, and MMP10, which is interesting and could have therapeutic implications. However, without actual animal or human experimental studies, the relevance of these findings to human TB are still very uncertain. As the authors would surely attest, human TB lesions are far more complex than can be represented by the models presented here. The purpose of these experiments is of course to lay the groundwork for future animal models and for eventual human trials if justified, and the authors have done an excellent job describing their findings. However, an inherent and notable limitation of the paper is the lack of experimental animal models targeting these mechanisms.

2) The conclusions in the discussion mentioning “TB”. The authors should revise their conclusions to address all mention of the proposed mechanisms’ relevance to “TB”, as this undoubtedly will imply human TB to most readers. With the exception of 5 slides from lymph nodes from patients with extrapulmonary TB, what they have studied is not “TB”, but in vitro and very limited in vivo (murine) models whose relevance to human patients with TB or at risk of TB is debatable. Some specific examples include:

a. Page 21. 1st paragraph.

“Extracellular acidosis and lactate thus independently… enhance secretion of MMP1 and its activators in TB…”

This should be revised to state that this was seen in these specific models, and the applicability of this to human TB needs further investigation. For example, since lactate is associated with acidosis in tissue, how could one ever discern that lactate and acidosis are independent of each other in human disease? While the findings may very well be relevant to human disease, the conclusions should stick to the data and anything else should be considered as suggestive or speculative, requiring further study.

b. Page 21. 1st paragraph.

“Consequently, acidosis and lactate exacerbated MMP-1 activity likely potentiates cavity formation and tissue destruction in TB which are key to disease transmission, treatment responsiveness and morbidity.”

This should be revised to state that the mechanisms whereby may be relevant to human TB, but this requires further study. While acidosis and Mtb infection clearly were synergistic here, no data are provided to indicate that this synergy is particularly important and targetable in the very complex lesions of human TB.

c. Page 22. 2nd paragraph

“TDAG8 is therefore the most significant acidosis sensing GPCR regulating macrophages responses in TB and pharmacological inhibition…”

This conclusion is not supported by the study of 5 patients with extrapulmonary TB and should be revised. A more appropriate conclusion focusing more on the findings and set in the context of the models used is needed.

3) (Minor) In the Discussion the authors seem to support a model where TNFa is uniformly beneficial to the host (Page 23, last paragraph). This is the case in patients who are latently infected and not on TB therapy, as illustrated by associations between TNFa modulators and TB reactivation. However, in patients on antimicrobials that treat TB, a major goal of treatment is in fact to decrease inflammation and TNFa levels. For example, the RCT by Mayanja-Kizza et al (JID 2005) specifically used a prednisolone dosage that was chosen based on its ability to reduce TNFa levels in whole blood cultures by 50%, and this dose then was associated with faster sputum culture conversion in adults with pulmonary TB. Given these RCT results, blocking TDAG-8 in order to increase TNFa levels in order to improve bacterial control is not an obvious approach in patients with pulmonary TB being treated with antitubercular therapy. Perhaps the most applicable statement would be that such an intervention might be useful in situations where TNFa deficiency is an issue thought to be increasing risk of TB reactivation. I suggest the authors are more specific in their vision of how this might be translated, even if to animal models, should they wish to propose this as a host-directed therapy.

**Part II – Major Issues: Key Experiments Required for Acceptance**

Reviewer #1: We are not told the source of H37RRv? It was grown in broth culture which creates clumps. An MOI of 1 was used. How was this determined and how was clumping assessed and dealt with?

Macrophages were cultivated in FCS which itself activates human cells. It is important to know whether cells cultivated in human AB serum provide the same result. Additionally, macrophages were cultivated in M-CSF which creates one subset of macrophages. What was the rationale for this approach?

Acidosis is known to affect phagocytosis (and C-type lectins were down). The authors should assess bacterial uptake under the various conditions of their assays.

The donor and pH grouping and size in Fig. 1A make it difficult to discern the finding of “demonstrated distinct transcription signatures for M.tb infection and acidosis”.

Fig. 5 D-L is missing 7.0 uninfected in the graphs. Although the protein data in Fig. 5 generally recapitulate the transcriptomic data for MMPs, most of the cytokine data were different. No adequate explanation was given. The authors conclude in the discussion that acidosis effects on cytokines is complex. Thus, the significance of the findings is in question.

Fig. 6A time point? Fig. 6 J: recovery was modest and from the methods section, the KD efficiency, esp. for OGR-1 was weak. Did the authors optimize the timing of KD for these experiments?

Figure 7: Overall cellular architecture is hard to visualize. It is surprising that theTDAG-8 is said to stain strongly in the necrotic center (although staining appears throughout to this reviewer). Necrotic centers usually lack intact macrophages. Is the visualized protein cell-associated in the necrotic center? Why not intense staining in the intermediate zone macrophages surrounding the necrotic center? I am not quite sure what I am looking at with the MGCs? Again, cellular architecture is not clear. 7F would benefit from arrows since I am not sure what we are looking at. Would the tissue cellular architecture information be improved by also staining NHP granulomas which allow for more defined cellular architecture? Several labs have available tissue and are willing to share.

Fig. 8. Based on the information, it is not clear what the conclusion is of the independent effects of lactate vs acidosis? Likewise, the MCT data are limited and confusing to interpret relative to the importance of lactate vs acidosis. The discussion mentions that lactate levels above 30mM are above that described in TB but no reference is given, and how and where this is measured is not mentioned. More clear discussion is needed on the overall interpretation of the results and what they mean.

Reviewer #2: none

Reviewer #3: See above.

**Part III – Minor Issues: Editorial and Data Presentation Modifications**

Reviewer #1: See above

Reviewer #2: 1. The authors should update their data on the number of annual TB deaths to the most recent report from 2022.

2. Bottom paragraph on p.5, MDMs not yet defined.

3. “Degradation” is misspelled in several Figures.

4. Although the methods indicate that Bonferroni corrections were used for some of the analyses, it is not clear in the results presentation when such corrections for multiple testing were actually employed (e.g., Fig 3).

Reviewer #3: See above.

PLOS authors have the option to publish the peer review history of their article (what does this mean?). If published, this will include your full peer review and any attached files.

Reviewer #1: No

Reviewer #2: No

Reviewer #3: No
---

## [Decision Letter · Decision Letter 1]

22 May 2023

Dear Dr. Whittington,

Thank you very much for submitting your manuscript "An acidic microenvironment in Tuberculosis increases extracellular matrix degradation by regulating macrophage inflammatory responses" for consideration at PLOS Pathogens. As with all papers reviewed by the journal, your manuscript was reviewed by members of the editorial board and by several independent reviewers. The reviewers appreciated the attention to an important topic. Based on the reviews, we are likely to accept this manuscript for publication, providing that you modify the manuscript according to the review recommendations.

Specifically, please note:

**Given the range of results and their complexity it is difficult to follow how the pathways described are integrated. This reviewer feels that a model figure that puts the findings together in a more cogent story would enhance the understanding by readership.**

Sincerely,

David M. Lewinsohn

Academic Editor

PLOS Pathogens

Marcel Behr

Section Editor

PLOS Pathogens

Kasturi Haldar

Editor-in-Chief

PLOS Pathogens

orcid.org/0000-0001-5065-158X

Michael Malim

Editor-in-Chief

PLOS Pathogens

orcid.org/0000-0002-7699-2064

Reviewer Comments (if any, and for reference):

Reviewer's Responses to Questions

**Part I - Summary**

Reviewer #1: The manuscript is improved as a result of editing, further explanation of results and some additional experimentation.

Reviewer #2: The revision has addressed previous critiques.

**Part II – Major Issues: Key Experiments Required for Acceptance**

Reviewer #1: (No Response)

Reviewer #2: (No Response)

**Part III – Minor Issues: Editorial and Data Presentation Modifications**

Reviewer #1: Given the range of results and their complexity it is difficult to follow how the pathways described are integrated. This reviewer feels that a model figure that puts the findings together in a more cogent story would enhance the understanding by readership.

Reviewer #2: (No Response)

PLOS authors have the option to publish the peer review history of their article (what does this mean?). If published, this will include your full peer review and any attached files.

Reviewer #1: No

Reviewer #2: No

Figure Files:

Data Requirements:

Reproducibility:

References:

---

## [Editor Report · Decision Letter 2]

20 Jun 2023

Dear Dr. Whittington,

We are pleased to inform you that your manuscript 'An acidic microenvironment in Tuberculosis increases extracellular matrix degradation by regulating macrophage inflammatory responses' has been provisionally accepted for publication in PLOS Pathogens.

Best regards,

David M. Lewinsohn

Academic Editor

PLOS Pathogens

Marcel Behr

Section Editor

PLOS Pathogens

Kasturi Haldar

Editor-in-Chief

PLOS Pathogens

orcid.org/0000-0001-5065-158X

Michael Malim

Editor-in-Chief

PLOS Pathogens

orcid.org/0000-0002-7699-2064
---

## [Editor Report · Acceptance letter]

4 Jul 2023

Dear Dr. Whittington,

We are delighted to inform you that your manuscript, "An acidic microenvironment in Tuberculosis increases extracellular matrix degradation by regulating macrophage inflammatory responses," has been formally accepted for publication in PLOS Pathogens.

Best regards,

Kasturi Haldar

Editor-in-Chief

PLOS Pathogens

orcid.org/0000-0001-5065-158X

Michael Malim

Editor-in-Chief

PLOS Pathogens

orcid.org/0000-0002-7699-2064